# Endothelial-to-Mesenchymal Transition in Health and Disease: Molecular Insights and Therapeutic Implications

**DOI:** 10.3390/ijms262311724

**Published:** 2025-12-03

**Authors:** Ran Kim, Woochul Chang

**Affiliations:** 1Department of Biology Education, College of Education, Pusan National University, Busan 46241, Republic of Korea; kimran2448@naver.com; 2Research Institute for Convergence of Biomedical Science and Technology, Pusan National University Yangsan Hospital, Yangsan 50611, Republic of Korea

**Keywords:** endothelial-to-mesenchymal transition, endothelial dysfunction, vascular remodeling, TGF-β/SMAD signaling, translational therapeutics

## Abstract

Endothelial-to-mesenchymal transition (EndMT) is a cellular program implicated in fibrosis, vascular remodeling, and the tumor microenvironment across multiple organs. We synthesize mechanistic pathways including TGF-β/SMAD, non-canonical (MAPK, PI3K/AKT, Rho/ROCK), Notch, and Wnt/β-catenin cascades. Their crosstalk with hypoxia, inflammatory cues, and epigenetic mechanisms can drive loss of endothelial identity and acquisition of mesenchymal characteristics. We outline disease contexts in the heart, lungs, kidneys, liver, central nervous system, and cancer, highlighting context-dependent contributory roles of EndMT. Therapeutically, we review pathway-targeted agents, epigenetic inhibitors, microRNA-based strategies, antibodies/biologics, small molecules and natural compounds, and cell- and gene-based interventions. Finally, we outline a translational roadmap that pairs patient-derived iPSC/organoid and organ-on-a-chip platforms to stratify EndMT states and prioritize targets. We also explore combination regimens that integrate multi-pathway modulation with epigenetic and immune approaches, aiming to deliver clinically meaningful anti-fibrotic benefits while better preserving physiological signaling.

## 1. Introduction

Endothelial cells (ECs) play a central role in the maintenance of vascular integrity and tissue homeostasis by sensing and integrating mechanical and biochemical stimuli [1]. They respond to shear stress, inflammatory cytokines, and oxidative stress, which affect the structural and functional integrity of the endothelial barrier and leukocyte–EC interactions [2,3,4,5]. ECs constitute a monolayer lining the luminal surface of blood vessels, serving as a dynamic barrier that regulates vascular tone, permeability, and interactions between blood components and surrounding tissues [6,7]. Disruption of endothelial homeostasis under pathophysiological conditions leads to endothelial dysfunction and endothelial-to-mesenchymal transition (EndMT) [8]. This process is implicated in vascular pathologies and is characterized by the loss of endothelial identity and the acquisition of mesenchymal traits such as cytoskeletal remodeling, cellular motility, and contractility [9]. Over the past decades, in vitro and in vivo studies have implicated endothelial phenotypic transition in pathological remodeling across multiple organs, including the cardiovascular system, lungs, kidneys, liver, and central nervous system [8,10,11,12,13,14]. Accumulating evidence indicates that EndMT contributes to disease processes such as cardiac fibrosis, pulmonary vascular remodeling/pulmonary arterial hypertension (PAH), kidney and liver fibrosis, cerebral cavernous malformations (CCMs), and context-dependent blood–brain barrier (BBB) dysfunction. Mapping of key EndMT pathways, including transforming growth factor-β (TGF-β)/SMAD, Notch, Wnt/β-catenin, and YAP/TAZ, has identified potential therapeutic targets, supported by preclinical and early-phase clinical studies [8,15,16,17]. Collectively, pathway mapping and target identification support the development of EndMT-targeted therapeutics for human diseases involving cellular transdifferentiation.

To enhance transparency and enable readers to assess the scope and potential sources of bias, we summarize below the literature search strategy and selection criteria used in this narrative review.

### Scope and Literature Selection

This review provides a narrative synthesis of EndMT in fibrosis, vascular remodeling, and the tumor microenvironment. We prioritized peer-reviewed primary studies and authoritative reviews searched in PubMed, Web of Science, and Scopus (databases last searched on 27 November 2025), using combinations of keywords including “endothelial-to-mesenchymal transition”, “EndMT”, “EndoMT”, “partial EndMT”, “fibrosis”, “vascular remodeling”, “barrier dysfunction”, “spatial transcriptomics”, “organoid”, and “organ-on-chip”. Additional relevant articles were identified through citation tracking of key papers. Inclusion criteria were as follows: (i) studies explicitly addressing EndMT (or endothelial mesenchymal activation/partial EndMT) in vascular or organ-specific endothelium; and/or (ii) studies providing mechanistic, methodological, or therapeutic modulation evidence relevant to EndMT. Evidence was categorized by experimental context (in vitro, in vivo, ex vivo, and human tissues/datasets). Exclusion criteria included articles not relevant to EndMT, studies lacking EndMT-related endpoints, and non-research items. Where cited, preprints were used sparingly to highlight emerging technologies and explicitly labeled as such.

## 2. Molecular Mechanisms and Pathophysiological Roles of EndMT

EndMT is orchestrated by various signaling pathways, including TGF-β/SMAD, Notch, Wnt/β-catenin, PI3K/AKT cascades, as well as inflammatory cytokines. It is governed by transcriptional networks comprising SNAI1, SNAI2, ZEB1/2, and TWIST1 [8,9,15,18,19,20,21]. Noncoding RNAs and epigenetic regulators also modulate the process [22,23,24]. Phenotypically, EndMT is commonly characterized by loss of endothelial markers (CD31, vWF, VE-cadherin) and gain of mesenchymal markers (α-SMA, fibronectin, vimentin) across developmental and pathological conditions [9,22,23]. EndMT is a subtype of epithelial-to-mesenchymal transition (EMT) that shares key regulatory pathways and transcriptional effectors [8,25]. It is essential for embryonic heart development and contributes to various pathological processes in postnatal tissues, including fibrosis, inflammation, and vascular remodeling [26,27]. EndMT is regulated by extrinsic cues (TGF-β, inflammatory cytokines), hypoxia, and oxidative stress [9,27]. The intrinsic transcriptional network also drives phenotypic transitions that are further shaped by epigenetic mechanisms, including DNA methylation, histone modifications, and microRNAs (miRs) [9,22,23,24] (Table 1).

### 2.1. Transforming Growth Factor-β

TGF-β signaling drives a wide range of organ-specific pathological processes across the lungs, kidneys, heart, and tumors [18]. It stimulates fibroblast-to-myofibroblast differentiation, extracellular matrix (ECM) deposition, pulmonary fibrosis and remodeling [28]. TGF-β induces tubulointerstitial and glomerular fibrosis through SMAD-dependent pathways in the kidneys [57]. TGF-β also facilitates post-infarcted and hypertrophic remodeling in myocardium and activates cancer-associated fibroblasts, which remodel the ECM and promote cancer progression and metastasis in the tumor microenvironment [58,59]. In ECs, TGF-β binds TGF-β receptor (TβR) II and activates TβRI to initiate the canonical SMAD2/3/4 signaling and non-canonical cascades [15,29]. In the canonical pathway, phosphorylated SMAD2/3 forms complexes with SMAD4 and translocates to the nucleus, where it induces transcriptional programs involving EndMT-associated transcription factors such as SNAI1, SNAI2, and TWIST [15,29]. In non-canonical signaling, p38 MAPK/JNK, PI3K/AKT, Rho GTPases, and NF-κB are engaged in cytoskeletal reorganization, cell motility and survival, and intersect with SMAD signaling to help coordinate the transcriptional reprogramming associated with EndMT initiation and progression [15,60].

### 2.2. Wnt/β-Catenin Signaling in EndMT

Canonical Wnt/β-catenin signaling has been implicated in EndMT, particularly in fibrotic and inflammatory contexts [25,61,62]. In several EndMT models, β-catenin-dependent transcription has been shown to attenuate endothelial gene programs and promote mesenchymal features, often accompanied by barrier dysfunction and profibrotic remodeling [51,63,64]. Mechanistically, Wnt ligands engage Frizzled and LRP5/6 co-receptors to stabilize β-catenin, allowing its nuclear translocation and transcriptional activation with TCF/LEF [65]. Genetic evidence also supports a requirement for endothelial β-catenin during developmental EndMT in cardiac cushion formation [66]. In the disease context, aberrant Wnt activation has been associated with EndMT-like programs in the infarcted heart and diabetic kidney disease [63,64,67]. Importantly, functional coupling between TGF-β and Wnt/β-catenin signaling has been reported in EndMT, indicating that their cross-talk may modulate the magnitude and persistence of EndMT responses [25,62].

### 2.3. Transcriptional Regulation of EndMT

Downstream transcription factors drive EndMT by repressing endothelial genes and simultaneously activating mesenchymal programs, whereas coupling to Rho GTPase family–dependent cytoskeletal remodeling enhances cell motility [9,60]. TGF-β signaling can upregulate MMP-2 and MMP-9, degrading the basement membrane/ECM and reinforcing EndMT progression in fibrosis [68,69]. Over the past decade, numerous studies have identified SNAI2, TWIST1 and ZEB1/2 as key transcription factors implicated in EndMT [9,25]. SNAI2 is essential for endocardial cushion morphogenesis during cardiac development, and implicated in repressing endothelial markers and promoting mesenchymal traits in disease contexts [19]. TWIST1 can be modulated in a context-dependent manner by developmental and mechanotransductive cues (Notch-dependent regulation, flow-responsive programs), and promotes EC motility and cytoskeletal remodeling in atherosclerosis [12,20,25,70]. ZEB transcription factors act as key drivers of EndMT by suppressing endothelial identity and activating mesenchymal programs [25]. The Notch pathway can modulate EndMT via Notch–SNAI2 in cardiac cushion morphogenesis or MMP-9-Notch signaling in kidney transdifferentiation [19,68]. Collectively, these mechanisms drive EndMT under pathological contexts.

### 2.4. Epigenetics

Epigenetics refers to the heritable changes in gene expression without alterations in the DNA sequence, primarily via DNA methylation, histone modifications, and noncoding RNA-mediated regulation [71]. These mechanisms have been implicated in modulating transcription factor networks that influence EndMT in developmental contexts (heart valve/endocardial cushion formation) and in pathological states such as cardiac fibrosis, diabetic cardiomyopathy, and vascular remodeling [8,15,47,72].

#### 2.4.1. DNA Methylation

DNA methylation is a key epigenetic mechanism regulating EndMT [24]. Under specific pathological cues, notably disturbed flow and inorganic phosphate, DNA methyltransferase 1 (DNMT1)-dependent promoter hypermethylation and gene silencing have been reported in ECs undergoing mesenchymal activation [24,36,43]. These changes can diminish endothelial identity programs, such as KLF4 locus methylation, and are accompanied by epigenetic repression, favoring EndMT-associated transcriptional states [36,37,73]. Pharmacologic DNMT inhibition or genetic targeting of DNMT1 mitigates EndMT features in cellular and vascular models [24,43].

#### 2.4.2. Histone Modification

The histone tails of the core octamers (H2A, H2B, H3, and H4) are the principal substrates for epigenetic modifications that control chromatin accessibility and gene expression in ECs [74]. Acetylation and methylation dynamically tune EndMT-related transcriptional programs [23]. Class I/II histone deacetylase (HDAC) activity, particularly HDAC9 and HDAC3, can promote EndMT and adverse vascular remodeling, whereas HDAC9 loss-of-function or pharmacologic HDAC3 inhibition attenuates EndMT in preclinical models [22,44,75]. The histone demethylase JMJD2B represses H3K9me3 at EndMT-associated promoters (SULF1 and CNN1) to enhance their expression [23]. In contrast, EZH2-mediated H3K27me3 has been linked to silencing of endothelial-maintaining or anti-fibrotic programs and facilitating EndMT and fibrogenic responses [45,46]. Together, pharmacological targeting of these enzymes can partially reverse mesenchymal reprogramming, suggesting that histone modification is a therapeutic candidate for EndMT-associated diseases.

#### 2.4.3. MicroRNA

miRs are short noncoding RNAs that repress gene expression post-transcriptionally by base-pairing with target mRNAs that bidirectionally control EndMT promotion or inhibition [46]. EndMT-relevant miRs can modulate TGF-β/SMAD signaling at multiple nodes, and in certain contexts, they also influence related pathways such as Wnt/β-catenin, biasing ECs toward either endothelial maintenance or mesenchymal activation programs [49,76,77,78,79,80]. MiR-20a targets TβRII and SARA to dampen canonical TGF-β signaling and inhibit EndMT [76]. Members of the miR-200 family, especially miR-200b, help preserve endothelial identity by blocking ZEB1/2 and EndMT-promoting pathways [48]. In contrast, miR-21, miR-27 and miR-155 can facilitate TGF-β-driven EndMT and vascular remodeling [49,77,78]. Notably, emerging evidence suggests context-dependent pro-EndMT roles for specific miRs, including miR-200c-3p [81,82]. In addition, miR-632 has been linked to EndMT activation and fibrosis in Marfan syndrome aortopathy [83].

Collectively, epigenetic and post-transcriptional mechanisms intersect with transcriptional programs to coordinate EndMT progression, and targeting these integrated networks offers a promising strategy for preventing or treating EndMT-mediated pathologies [22,23,24,72].

### 2.5. Plasticity and Reversibility of EndMT

EndMT often manifests as a spectrum, including partial or intermediate states, and its apparent reversibility is highly context-dependent [15]. In vitro, induction of withdrawal paradigms and reinforcement of pro-endothelial pathways (KLF2/4, BMP7, and NO signaling) can partially restore endothelial markers and reduce mesenchymal features [84,85,86]. In vivo, endothelial lineage-tracing and time-resolved disease models support an endothelial contribution to mesenchymal-like populations [87]. However, evidence for sustained reversibility remains model- and readout-dependent and can be confounded by Cre-driver specificity/efficiency and marker overlap across stromal lineages [88,89,90]. Ex vivo/human tissues and single-cell/spatial datasets largely provide cross-sectional snapshots of mixed endothelial/mesenchymal programs and inferred trajectories rather than direct state transitions, underscoring the need for longitudinal sampling and lineage-aware, orthogonal validation when feasible [91,92,93].

Taken together, these signaling and epigenetic mechanisms provided a shared mechanistic framework that is deployed in an organ- and disease context-dependent manner [8,9,15]. In the following section, we discuss how canonical TGF-β/SMAD and non-canonical MAPK and PI3K/AKT cascades, Notch and Wnt/β-catenin signaling, hypoxia/HIF-driven responses, and epigenetic modifiers such as DNMTs, HDACs, and histone methyltransferases converge to drive EndMT in specific organ systems [15,24,25,29,60]. We also highlight how differences in hemodynamic load, local inflammatory milieu, and metabolic stress bias the contribution of these pathways to EndMT in the cardiovascular system, lungs, kidneys, liver, central nervous system, and tumor microenvironment [2,7,10,11,12,13,69] (Figure 1).

## 3. Organ System Pathologies Associated with EndMT

EndMT, which is recognized as a developmental program during endocardial cushion formation and valvulogenesis, has emerged as a disease-related mechanism in adult organs [8]. It contributes to atherosclerosis and vascular remodeling of the cardiovascular system, pulmonary hypertension, lung remodeling, and renal fibrosis via EndMT programs [8,9]. It can also contribute to tumor progression, partly through endothelial-derived carcinoma-associated fibroblasts, CCMs in the nervous system and hepatic pathology [9,94,95]. Collectively, these studies support EndMT as a mechanistic contributor to fibrogenesis, chronic inflammation, and tissue remodeling in multiple organ systems.

### 3.1. Cardiovascular System

EndMT is essential during embryogenesis and is increasingly recognized as a contributor to adult cardiovascular diseases, including cardiac fibrosis, valvular disorders, and atherosclerosis [8,96,97]. ECs undergo EndMT and generate fibroblast-like or myofibroblast populations that secrete the ECM, increasing collagen deposition and ventricular stiffening [8,30,98]. Lineage-tracing and animal studies indicate that a subset of cardiac fibroblasts derives from ECs via EndMT [30,87,98]. Moreover, the experimental suppression of EndMT mitigates interstitial and perivascular fibrosis after MI, linking EndMT to adverse ventricular remodeling and diastolic dysfunction [8,50,99].

Valvular ECs also undergo EndMT in calcific aortic valve disease, which contributes to fibrotic and calcific remodeling of stenotic valves [31,96]. Human calcified aortic valve and ApoE−/− mouse valves have reported cell populations co-expressing endothelial and mesenchymal markers, consistent with in situ EndMT [31]. Pro-calcific or pro-fibrotic cues, including TGF-β, BMP, and Notch, modulated by inflammatory and biomechanical stress, can induce EndMT in valvular ECs, supporting EndMT as a mechanistic contributor and potential therapeutic target in valvular remodeling [32,100,101].

In atherosclerosis, ECs located in regions exposed to disturbed flow exhibit EndMT-like programs, losing their endothelial identity and acquiring mesenchymal traits [43]. This transition is associated with increased ECM production and pro-inflammatory signaling within plaques, which contribute to the feature of plaque complexity and vulnerability [22,102]. Collectively, these findings underscore EndMT as a pathogenic contributor to vascular remodeling and highlight it as a potential therapeutic target in atherosclerotic disease.

Mechanistically, cardiovascular EndMT is driven predominantly by TGF-β/SMAD signaling and its non-canonical branches (p38/JNK, PI3K/AKT, and RhoA/ROCK), which are activated by pressure overload, neurohumoral activation, and inflammatory cytokines [15,103,104,105]. Disturbed flow at an atheroprone site further engages Notch signaling, TWIST1, and DNMT1-dependent epigenetic reprogramming, while hypoxia and oxidative stress stabilize mesenchymal gene programs [36,37,40,43]. These coordinated inputs link hemodynamic and inflammatory stress to endothelial plasticity, fibroblast accumulation, and adverse cardiac and vascular remodeling [7,8,30].

### 3.2. Lung

In idiopathic pulmonary fibrosis (IPF), lung ECs exposed to chronic injury and pro-fibrotic stimuli (TGF-β, hypoxia) have been reported to undergo EndMT, contributing to the emergence of fibroblast-like or myofibroblast populations, excessive ECM deposition and interstitial scarring that disrupts the alveolar-capillary interface and impairs gas exchange [106,107,108]. In PAH, pulmonary ECs subjected to disturbed shear and inflammatory cytokines exhibit EndMT features, promoting intimal thickening and vascular remodeling that elevate pulmonary vascular resistance and pressures, with downstream right ventricular strain [10,109]. TGF-β/SMAD, BMP and PDGF cascades intersect with biomechanical stress to influence EndMT, highlighting these pathways as potential therapeutic targets under investigation in chronic lung disease [10].

In these pulmonary settings, EndMT is orchestrated mainly by TGF-β/SMAD and BMP signaling in conjunction with hypoxia/HIF-1α-TWIST1 pathways and PDGF-driven proliferative cues, which together promote the transition of pulmonary ECs toward a mesenchymal and pro-fibrotic phenotype [10,40,107].

### 3.3. Kidney

EndMT contributes to renal fibrosis in chronic kidney disease (CKD), as lineage-tracing has shown that a subset of renal fibroblasts derives from ECs and is associated with microvascular rarefaction and disease progression [110,111]. The TGF-β/SMAD pathway, along with non-canonical cascades (ERK, p38 MAPK, PI3K/AKT), and Wnt/β-catenin and Notch signaling, can drive renal EndMT [40,61,86]. These pathways are activated by angiotensin II, oxidative stress/ROS and hyperglycemia, as well as inflammatory cytokines (IL-1β, TNF-α) [112]. Under chronic hypoxic conditions, endothelial HIF-dependent signaling may promote EndMT and microvascular rarefaction, which worsens oxygen delivery and causes peritubular injury, accelerating renal remodeling in CKD [111]. Thus, renal EndMT exemplifies how canonical TGF-β/SMAD, Wnt/β-catenin, and Notch signailing intersect with metabolic and oxidative stress to drive microvascular rarefaction and progressive tubulointerstitial fibrosis [11,57,68,110,112,113].

### 3.4. Liver

In chronic liver disease, liver sinusoidal ECs undergo capillarization and can adopt partial EndMT features under pro-fibrotic cues (TGF-β, hypoxia), contributing to excess ECM deposition and fibrotic remodeling in experimental in vivo models [114]. Ex vivo analyses of human fibrotic liver tissues/datasets further report endothelial populations with mixed endothelial and mesenchymal signatures consistent with partial EndMT [115]. Although stellate cells remain the principal source of myofibroblasts, context-dependent liver sinusoidal EC-EndMT may augment fibrogenesis and vascular remodeling as the disease advances [114,116]. Available evidence suggests that sinusoidal EndMT in chronic liver disease in driven largely by TGF-β/SMAD and hypoxia/HIF pathways, potentially modulated by inflammatory cytokines and epigenetic changes, amplifying fibrogenic and vascular remodeling responses as disease progresses [12,114,117,118].

### 3.5. Central Nervous System

In the central nervous system (CNS), neuroinflammatory cues (TGF-β, IL-1β) can induce EndMT in brain microvascular ECs, contributing to loss of BBB integrity and enhanced leukocyte infiltration in multiple sclerosis and experimental autoimmune encephalomyelitis [13,116]. Regulators, such as ETS1, modulate EndMT and BBB integrity. Inhibition of ETS1 has been associated with EndMT-linked BBB dysfunction in these models [13]. Genetic and cellular evidence from cerebral cavernous malformation analyses further supports a pathogenic role of EndMT in CNS vascular dysfunction [95]. At the molecular level, CNS EndMT appears to be governed by TGF-β/SMAD signaling together with inflammatory pathways such as NF-κB and transcriptional regulators, including ETS1, linking neuroinflammation and BBB breakdown to endothelial plasticity [13,119].

### 3.6. Tumor Microenvironment

In cancer, ECs undergoing EndMT lose endothelial markers and acquire mesenchymal traits, contributing to a subset of cancer-associated fibroblasts and secreting cytokines/ECM components that can promote tumor progression, angiogenesis, and immune evasion [94]. This phenomenon has been documented in breast, colon, and pancreatic cancers and is linked to inflammatory cues and TGF-β-driven EndMT programs [69,119,120]. Tumor-associated EndMT is primarily driven by TGF-β/SMAD signaling in concert with inflammatory cytokines and pro-angiogenic cues, accompanied by transcriptional and epigenetic reprogramming that generates EndMT-derived cancer-associated fibroblasts and reinforces an immune-evasive, pro-fibrotic tumor microenvironment [69,94,120,121].

## 4. Disease Models and Assessment Methods for EndMT

Refined in vitro (TGF-β, hypoxia, 2D-3D matrices, microfluidic shear) and in vivo (endothelial lineage-tracing, disease-specific) models with standardized marker panels, functional readouts and single-cell sequencing have facilitated EndMT research and informed therapeutic discovery across vascular and fibrotic diseases [37,91,122,123,124] (Table 2).

### 4.1. In Vitro Model Systems

#### 4.1.1. Mechanochemical Stimulation-Based Models

In vivo, ECs are consistently exposed to dynamic mechanical forces, including shear stress, cyclic stretch, and ECM-derived cues that help maintain vascular homeostasis [38,136]. However, disturbed or pathological mechanics can reprogram endothelial phenotype and promote EndMT by engaging pathways such as Notch and TGF-β, with downstream effectors including SNAI1 and TWIST, while cell–matrix traction and substrate stiffness regulate barrier function and transcriptional states [38,136]. In vitro models using oscillatory-flow systems, hypoxic chambers, and stiff substrates have shown that disturbed shear and hypoxia can induce EndMT via Notch- and HIF-dependent pathways, whereas pathological matrix stiffness drives ECs toward the mesenchymal state [16,40,127]. These platforms provide mechanistic insight into how aberrant mechanical environments promote endothelial dysfunction and EndMT in vitro.

#### 4.1.2. Two-Dimensional Monolayer Culture Models

Two-dimensional monolayer cultures of primary ECs (HUVECs, HAECs) are widely used platforms for studying EndMT, because they enable tightly controlled stimulation with defined inducers including TGF-β, TNF-α and hypoxia [16,125,137]. These models are convenient and reproducible; however, they lack the spatial and cellular heterogeneity of native tissues and microenvironments [126]. After EndMT induction, progression is quantified by shifts in endothelial/mesenchymal marker expression, barrier readouts such as transendothelial electrical resistance or FITC-dextran permeability, and morphological/cytoskeletal remodeling [13,38,40]. To overcome 2D limitations, many studies combine hypoxia or controlled flow with stiff substrates and transition to 3D and microfluidic systems that better recapitulate endothelial mechanochemical conditions [16,127,131,132].

#### 4.1.3. Three-Dimensional Models and Organoid Systems

Three-dimensional collagen or Matrigel cultures are suitable for direct studies of cell–matrix interactions and structural remodeling during EndMT [128]. More advanced platforms, such as iPSC-derived vascular organoids and microfluidic organ-on-a-chip systems, incorporate physical cues (shear stress and hypoxia) to better approximate in vivo environments and provide more physiologically relevant EndMT dynamics [129,132]. In these systems, EndMT is tracked by dual-marker shifts, functional barrier readouts, and collagen deposition, including via second-harmonic generation imaging [138]. Organoids are valuable for capturing multicellular crosstalk and hypoxic gradients, while microfluidic chips enable precise control of shear profiles and real-time readouts under live imaging [130].

However, these platforms can exhibit batch-to-batch variability and differentiation-dependent heterogeneity in cellular composition and microenvironmental parameters [129,130,132]. Therefore, reproducibility is improved by standardized differentiation/operating protocols and transparent reporting of core QC metrics alongside EndMT readouts [129,130,132].

#### 4.1.4. Hypoxia-Based In Vitro Models

Coronary microvascular rarefaction exacerbates tissue hypoxia and is associated with cardiac fibrosis and adverse ventricular remodeling [41,111,139]. Pathological observations have shown microvascular rarefaction and dysfunction with myocardial fibrosis and left ventricular remodeling, while hypoxia/HIF-1α signaling is implicated in these processes [41,111,139]. Hypoxia can induce EndMT via HIF-1α/TWIST1 and TGF-β/SMAD pathways, mediating the phenotypic transitions reported in PAH and cardiac fibrosis and supporting hypoxic ECs culture as an informative in vitro platform for EndMT research [16,42]. In PAH models, hypoxia/HIF-1α signaling engages TWIST1/PDGFB and TGF-β/SMAD, which promote intimal thickening and arterial remodeling, supporting these pathways as potential therapeutic targets [42,109].

### 4.2. In Vivo Model Systems

Genetically engineered mouse models for EndMT use endothelial lineage-specific Cre drivers (Tie2-Cre, VE-cadherin-CreERT2) with Rosa26 reporters to trace endothelial progeny [133,134,135]. Complementary functional genetics, such as Cre-induced TβRII deletion and SMAD3 deficiency or perturbation, allow tests of necessity across canonical fibrosis models, including MI, unilateral ureteral obstruction, and bleomycin-induced pulmonary fibrosis [57,140,141,142]. Zebrafish, with optical transparency and rapid development, enable real-time vascular-/fibrosis-related imaging using transgenic reporters and live biosensors [143,144]. However, because injured zebrafish often display transient or limited fibrotic scarring due to robust regeneration, caution is warranted when translating findings to persistent mammalian fibrosis [145]. In contrast, rat models can offer closer physiological concordance for organ-level fibrotic outcomes and are suitable for multimodal readouts, including MRI and PET/CT [146].

### 4.3. EndMT Detection and Quantification Methods

#### 4.3.1. Marker-Panels and 3D Tissue Imaging

Marker-based EndMT detection typically relies on identifying loss of endothelial markers alongside gain of mesenchymal markers, quantified by immunofluorescence, flow cytometry, and immunoblotting [9,122]. While single markers are neither necessary nor sufficient to establish EndMT progression, and some fibroblast markers lack specificity, dual or multiplex panels are recommended to reduce misclassification and to support phenotypic calls within biologically coherent programs [8,88]. 3D tissue-level assessment of EndMT within fibrotic niches can be achieved with tissue clearing and light-sheet microscopy [14]. When these approaches are combined with appropriate lineage reporters, they enable whole-organ visualization of endothelial progeny and spatial quantification of their positions and fractions relative to collagen-dense scars [14,89].

In practice, the sensitivity of imaging-based detection depends on antibody validation and signal-to-noise, and on whole-mount immunolabeling performance and clearing/optional conditions [8,14,88,89]. Specificity is limited by marker overlap across endothelial, mural, and fibroblast-like lineages. Therefore, EndMT calls should be supported by coherent multi-marker programs rather than single markers [8,9,15,88]. To improve inter-laboratory reproducibility, studies should report the exact marker panel/antibody clones, clearing and imaging parameters, and quantification thresholds, and confirm key findings using an orthogonal modality when feasible [89].

#### 4.3.2. EndMT Fate Mapping and Transcriptional Resolution

In vivo lineage tracing of EndMT commonly employs endothelial-specific Cre drivers crossed with Rosa26 reporters, providing permanent genetic labeling of endothelial progeny and supporting an endothelial contribution [133,134,135]. As recombination efficiency and specificity depend on the induction timing/dose and controls, best practice includes dual-recombinase intersectional strategies to increase lineage specificity [89,147]. Ex vivo single-cell transcriptional approaches use scRNA-seq to resolve endothelial subpopulations and transcriptional states during EndMT progression [124]. Integration of scRNA-seq with scATAC-seq has also identified motif activity inferred from chromatin accessibility [37]. Spatial transcriptomics has localized signatures consistent with EndMT programs within organ-specific niches (post-myocardial infarction (MI) heart, IPF lungs, atherosclerotic plaques), enabling spatial mapping of putative transitions from endothelial identity loss to mesenchymal acquisition [91,102,148].

For sequencing-based readouts, sensitivity for rare or transient EndMT states is influenced by sampling depth and transcript dropout, while specificity can be affected by dissociation-induced transcript dropout, ambient RNA contamination, and cell doublets that blur endothelial–mesenchymal boundaries [149,150]. Accordingly, trajectory outputs should be interpreted as inference and anchored to phenotypic/lineage-aware validation where possible [92]. Finally, cross-study comparability requires reproducibility safeguards, including clear reporting of tissue processing, QC metrics, batch-aware computational workflows, and clearly defined EndMT signature criteria/thresholds [149,151].

### 4.4. Organ-Specific Considerations for EndMT Models and Assessment

From a practical standpoint, EndMT research typically combines a disease-relevant in vivo model with a standardized detection workflow (Section 4.1, Section 4.2 and Section 4.3). In the cardiovascular studies, MI or pressure-overload models are commonly used to link EndMT signatures to fibrotic remodeling, whereas atherosclerosis models under disturbed-flow conditions are used to examine plaque-associated EndMT [22,30,43]. In pulmonary diseases, bleomycin-induced fibrosis and hypoxia-based pulmonary hypertension models are frequently employed to assess EndMT-associated vascular remodeling using multi-marker staining and morphometric readouts [40,54,142]. Renal EndMT is often evaluated in unilateral ureteral obstruction and diabetic or hypertensive nephropathy models, alongside endpoints such as collagen deposition and microvascular rarefaction [57,63,110]. In more context-dependent settings (liver, CNS, and cancer), chronic injury or inflammatory disease models combined with multi-marker staining and, where applicable, single-cell or spatial profiling, are increasingly used to localize EndMT-like states within disease niches [88,91,114,120,127].

## 5. Therapeutic Regulation of EndMT

### 5.1. Signaling Pathway Inhibitors

#### 5.1.1. Inhibition of the TGF-β Pathway

The TGF-β pathway is a central and extensively studied driver of EndMT. Pharmacologic blockade of TβRI with small molecules such as SB-431542 suppresses TGF-β-induced EndMT by preventing SMAD2/3 phosphorylation in vitro and in disease contexts [33,152]. A-83-01, another TβRI inhibitor, similarly suppresses TGF-β-driven transdifferentiation [34]. Galunisertib (LY2157299), an oral TβRI inhibitor, has shown anti-fibrotic/anti-tumor activity in preclinical studies and early clinical signals in hepatocellular carcinoma and pancreatic cancer trials [153,154]. Because biomarker-guided patient selection has emerged as important in clinical experiences with TGF-β pathway inhibition, integrating pharmacodynamic markers into study design is advisable [153]. In addition, TGF-β signaling engages non-canonical branches (p38/JNK/ERK, RhoA/ROCK, PI3K/AKT) in EndMT, pairing TβRI blockade with barrier-function readouts, cytoskeletal remodeling, and transcriptional programs is commonly employed [29,60].

#### 5.1.2. Modulation of Notch Signaling

Notch signaling, which is activated through juxtacrine interactions, is implicated in EndMT in developmental and pathological contexts, including atherosclerosis, cardiac fibrosis, and vascular malformations [27,155]. Pharmacologic γ-secretase inhibition with DAPT, a Notch pathway inhibitor, has been reported to attenuate TGF-β-driven EndMT in endothelial models [156]. Disturbed flow-activated JAG1-NOTCH4 promotes vascular dysfunction and EndMT-like programs in atherosclerosis, indicating that Notch1 and Notch4 are context-dependent regulators of endothelial fate [39]. Given that broad γ-secretase/Notch blockade has raised concerns about endothelial homeostasis and dose-limiting toxicities, context-selective targeting of disturbed-flow-activated JAG1-NOTCH4 has been proposed as a strategy that may suppress pathological EndMT while better preserving physiological Notch signaling [38,39].

#### 5.1.3. Modulation of Wnt/β-Catenin Pathway

Wnt/β-catenin signaling is an important regulator implicated in EndMT, particularly in fibrotic and inflammatory conditions [25]. Pharmacologic disruption of the β-catenin/CBP interaction with ICG-001, a selective β-catenin/CBP inhibitor, dampens mesenchymal gene programs and can partially restore endothelial features [51,52]. Notably, β-catenin inhibition rescues AKT1-suppression-driven EndMT and vascular remodeling in vivo [51]. PRI-724, a second-generation β-catenin/CBP antagonist, shares the same mechanism as ICG-001 and supports a strategy to modulate β-catenin/CBP-dependent transcription without broadly shutting down Wnt signaling [53].

### 5.2. Epigenetic and ncRNA-Based Therapy Strategies

#### 5.2.1. Epigenetic Inhibitors: HDAC and DNMT Inhibition

HDACs and DNMTs are key epigenetic regulators of EndMT [10,22]. Activity of HDAC9 and HDAC3 has been implicated in promoting mesenchymal reprogramming during EndMT, while HDAC inhibition has been reported to restore histone acetylation and attenuate TGF-β-driven EndMT features [22,44,157]. Conversely, disturbed flow can induce DNMT1-dependent hypermethylation at mechanosensitive promoters, such as KLF4 [36,43]. DNMT inhibition with 5-aza-2′-deoxycytidine has been reported to attenuate EndMT and endothelial dysfunction in preclinical models [10].

#### 5.2.2. microRNA-Based Therapeutics

The miR-200 family (miR-200a/-200b/-429) suppresses EndMT by directly targeting ZEB1/2 and its loss under fibrotic or inflammatory stress promotes mesenchymal reprogramming [48,72]. Conversely, miR-21 is upregulated by TGF-β and accelerates EndMT via SMAD7 repression and AKT signaling [49,50]. Restoring anti-EndMT miRs, such as miR-200b, or inhibiting miR-21 attenuates mesenchymal marker expression and fibrosis in preclinical models [47,158]. Exosomes, lipid nanoparticles, and polymeric carriers can improve the stability and tissue targeting of miR therapeutics [159,160,161]. Endothelial-trophic delivery strategies underscore the importance of cell-specific targeting to minimize off-target pathway perturbations [162].

### 5.3. Antibodies and Biologic Agents

Monoclonal antibodies that modulate TGF-β signaling have shown preclinical and early clinical signals against fibrosis and EndMT-associated pathologies [35]. Fresolimumab, a pan-TGF-β-neutralizing antibody, suppresses TGF-β-driven fibrotic programs and showed clinical activity in early systemic sclerosis studies [163]. The anti-endoglin antibody TRC105 antagonizes endoglin co-receptor activity, ameliorates endothelial dysfunction, and targets pathological angiogenesis, consistent with the context-dependent role of endoglin in EndMT and vascular remodeling [164,165]. Because broad TGF-β blockade can cause dose-limiting toxicities, isoform- or co-receptor-targeted strategies, such as endoglin-directed modulation, have been proposed to better suppress pathological EndMT while sparing physiological signaling [17,166].

### 5.4. Small Molecule- and Compound-Based Strategies

Small molecule- and compound-based approaches have been reported to modulate pathological EndMT under fibrotic conditions [167]. Pirfenidone preserves endothelial network integrity and attenuates mesenchymal programs, in part via inhibition of Rho-kinase activity [55]. Nintedanib suppresses EndMT by reducing FAK activity and improves bleomycin-induced pulmonary fibrosis and experimental PAH [54,56]. Polyphenols, such as resveratrol and curcumin, have been shown to suppress EndMT/Endothelial-interstitial transitions in endothelial models [168,169]. Resveratrol counters NOX-mediated EndMT via PKC inhibition under high-glucose conditions, and curcumin reduces endothelial-interstitial transformation [168,169]. Combination strategies with anti-TGF-β or anti-inflammatory agents have been proposed to enhance anti-EndMT efficacy while limiting off-target effects [96,167].

### 5.5. Cell- and Gene-Based Therapeutic Strategies

iPSC-derived ECs (iPSC-ECs) have shown promise in preclinical models of vascular repair and attenuation of fibrotic remodeling [170,171]. CRISPR/Cas9 editing is feasible in primary ECs and iPSC-EC systems and provides a platform to modulate EndMT-relevant pathways in experimental settings [172,173]. Mechanistically, the miR-200 family suppresses EndMT, consistent with its direct repression of ZEB1/2 described in foundational EMT studies and validated in endothelial contexts [48]. In renal models, endothelial EndMT compromises vascular integrity and drives fibrosis via Myc-linked metabolic reprogramming, underscoring transcriptional regulators, such as SNAI1, as potential nodes for intervention and supporting cell-/gene-based strategies to stabilize engineered ECs [174]. Strategic vector selection, rigorous off-target profiling, and endothelial-tropic delivery are pivotal for translational success [162]. PECAM-1-targeting enables ApoE-independent lung-endothelial delivery of mRNA-loaded lipid nanoparticles, while PECAM-1/VCAM-1-directed DNA-lipid nanoparticles enhance the magnitude, duration, and organ specificity of endothelial transgene expression [162,175] (Table 3).

## 6. Conclusions

EndMT is increasingly recognized as an important contributor to fibrosis, vascular remodeling, and the tumor microenvironment [30,176]. The process appears organ- and time-dependent, creating opportunities for selective therapeutic interventions [7,124]. Recent advances in ex vivo single-cell omics have enabled the high-resolution profiling of EndMT-related populations across tissues and disease states, helping resolve cellular heterogeneity and infer transient versus more stable transition states [37,124]. When integrated with multiplexed and spatial imaging, these datasets can localize EndMT populations within tissue niches and map their interactions with stromal and immune cells, strengthening pathophysiological interpretation [91,102,148]. Additionally, patient-derived iPSC models and organoid/organ-on-a-chip systems may help establish disease-specific EndMT progression, supporting target discovery and exploration of patient-to-patient variability in drug responses [177,178,179]. However, translating findings from reductionist in vitro systems or emerging vascular organoid/on-chip platforms to human physiology remains challenging [126,130,132]. Key limitations include incomplete endothelial subtype maturation, reduced immune-stromal complexity, and cross-platform variability that complicates benchmarked endpoint alignment [129,132,180]. Finally, therapeutic strategies that combine pathway-directed agents with epigenetic modulation and immunoregulation may offer broader control of EndMT in preclinical models than single-pathway blockade, with potential to translate into meaningful anti-fibrotic effects [18,44,122].

Despite these advances, several conceptual and technical challenges remain. In vivo evidence often relies on endothelial lineage tracing and marker-based definitions [8,108,109,110]. However, lineage-tracing readouts can be confounded by Cre-driver recombination specificity/efficiency and induction timing and dose [109,110]. Marker-based definitions are limited by overlap and imperfect specificity of commonly used endothelial and fibroblast markers across related stromal lineages [8,108]. Single-cell and spatial datasets also provide largely static snapshots that infer, rather than directly observe, real-time state transitions [40,112]. Accordingly, integrating lineage information with longitudinal sampling and orthogonal functional readouts will be essential to distinguish EndMT trajectories from stable mesenchymal or perivascular phenotypes [31,109,110]. Key unresolved questions include the quantitative contribution of EndMT to fibroblast-like pools across organs and stages, whether partial EndMT reflects stable fate change versus adaptive endothelial activation, and the extent of reversibility in vivo under physiologically relevant conditions [15,30,181].

Most EndMT-targeted interventions remain at the preclinical stage, and safety, specificity, and delivery constraints continue to limit clinical translation [167,182]. While broad TGF-β pathway inhibition can attenuate EndMT and fibrosis, it is constrained by dose-limiting toxicities and the essential physiological roles of TGF-β signaling [35,183]. Likewise, miR- and gene-based approaches require rigorous off-target profiling and endothelial-tropic delivery to achieve durable and safe control in vivo [160,161,162,182]. Given the redundancy and extensive cross-talk among EndMT-regulatory networks, translational development will require mechanism-aware, context-selective strategies rather than single-pathway blockade [8,9,15]. Single-cell and spatial profiling can nominate tissue- and stage-resolved, program-level EndMT signature and trajectory-informed biomarker panels that support patient/lesion stratification and pharmacodynamic monitoring [91,93,124]. However, because snapshot omics infer rather than directly observe dynamic transitions, these signatures should be corroborated by orthogonal validation frameworks to establish in vivo dynamics under physiologically relevant conditions [89,92,144].

## Figures and Tables

**Figure 1 ijms-26-11724-f001:**
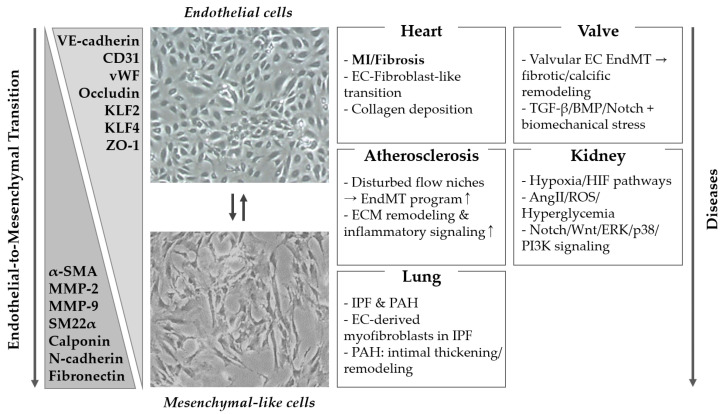
Overview of EndMT from endothelial identity to mesenchymal-like state across organs. **Abbreviations**: EndMT, endothelial-to-mesenchymal transition; EC, endothelial cell; MI, myocardial infarction; ECM, extracellular matrix; IPF, idiopathic pulmonary fibrosis; PAH, pulmonary arterial hypertension; HIF, hypoxia-inducible factor; AngII, angiotensin II; ROS, reactive oxygen species. **Symbols/arrows**: ↑ indicate increased expression/activity; → indicate “leads to/induces”. The central downward/upward arrow indicates progression from an endotelilal to a mesenchymal-like phenotype during EndMT and, where applicable, phenotypic plasticity/partial reversion toward an endothelial-like state, respectively.

**Table 1 ijms-26-11724-t001:** EndMT inducers, pathways, and disease contexts.

Inducer/ Stimulus	Major Pathway	Key TFs/ Regulators	Core Readouts	Disease Context	References
TGF-β	TβR1→SMAD2/3; PI3K/AKT, p38/JNK	SNAI1/2 TWIST1 ZEB1/2	↓CD31/VE-cadherin; ↑α-SMA, FN1, COL1A1	Cardiac fibrosis and valvular disease, aortic aneurysms, renal and pulmonary fibrosis	[8,15,28,29,30,31,32,33,34,35]
TNF-α/IL-1β	NF-κB, MAPKs	NF-κB (p65)	Barrier loss (TEER↓), Leukocyte adhesion↑; Synergizes with TGF-β to promote EndMT	Inflammatory vasculopathies, atherosclerosis	[3,5,7]
Disturbed/ oscillatory shear	Notch1, JAG-NOTCH4; DNMT1-mediated KLF4 promoter methylation	TWIST1 NOTCH1/4 DNMT1	EndMT program↑ EC identity↓	Atherosclerosis	[2,20,36,37,38,39]
Hypoxia	HIF-1α→TIWST1-PDGFB axis	HIF-1α TWIST1	EndMT markers↑ (α-SMA, FN1, SNAI1/2)	PAH, lung remodeling	[10,40,41,42]
Epigenetic regulation	HDAC9/HDAC3;DNMT1; JMJD2B;EZH2/H3K27me3	HDAC9/3 DNMT1 JMJD2B EZH2	H3K27me3 changes, KLF4 promoter methylation↑, miR-29c silencing	Atherosclerosis, Neointima, Fibrosis	[22,23,24,43,44,45,46]
miR-200 family	Direct targeting of ZEB1/2→EMT/EndMT suppression	miR-200a/b/c	ZEB1/2↓, Maintenance of EC Identity	Diabetic complications, Fibrosis	[47,48]
miR-21	Represses SMAD inhibitor→strengthens TGF-β	miR-21	EndMT markers↑, anti-miR-21 attenuates EndMT	Perivascular/ Cardiac fibrosis	[49,50]
Wnt/β-catenin (Therapy)	Inhibition of β-catenin/CBP transcriptional complex	-	EndMT suppression, Improved vascular Remodeling	Pulmonary fibrosis, Vasculopathy	[51,52,53]
Small molecules/ natural products(Therapy)	Rho-kinase/FAK↓	-	EndMT↓, Amelioration of fibrosis/PAH	Pulmonary fibrosis, PAH	[54,55,56]

Note: Arrows indicate direction of change relative to baseline ECs (↑up-regulated/increased; ↓down-regulated/decreased). The “→” symbol denoted a directional signaling/cause-effect relationship (activates/induces/leads to) between components. “-” is not applicable. Listed references are representative rather than exhaustive and were selected based on the inclusion/exclusion criteria.

**Table 2 ijms-26-11724-t002:** Key methodological platform for EndMT studies: strengths, limitations, and standardization considerations.

Platform	Major Strengths	Key Limitations/Pitfalls	Minimal Reporting/ QC Items	References
2D EC monolayers	High control, scalable, Convenient/reproducible Baseline	Limited tissue context Over-simplify transient/partial states	Cell source/passage; inducer dose/time; marker panel; replicate design	[125,126]
Mechnochemicalmodels	Mimics shear/stiffness- driven programs	Device-to-device variability; sensitivity to setup; lab-to-lab reproducibility issues	Flow/shear parameters; substrate stiffness; calibration method	[38,127]
2D ECM	Cell–matrix interaction, Structural remodeling	Matrix batch variability; limited cellular heterogeneity	Matrix composition/concentration; gel protocol; imaging quantification	[126,128]
iPSC-vascular organoids	Multicellular crosstalk, More physiological Gradients	Batch-to-batch variability; maturation state affects specificity	Differentiation QC; cell composition metrics; batch controls	[129,130]
Organ-on-a-chip/microfluidics	Controlled shear+ Real-time readouts	Fabrication/operation variability; throughput constraints	Device specs; shear profiles; barrier readouts; standardized operating protocol	[131,132]
In vivo disease models	System-level context, Causality testing	Species differences; model-specific confounders	Model details; time-course; endpoints; randomization/blinding if used	[125]
Endothelial lineagetracing	Lineage-informed “contribution” evidence	Recombination specificity/efficiency; time/dose confounds	Driver line; induction regimen; recombination efficiency controls	[133,134,135]
Marker panels	Widely accessible; quantitative protein-level calls	Specificity limited by marker overlap; single-marker false positives	Define “program-level” criteria; recombination efficiency controls	[8,9,88]
Tissue cleaning	Whole-organ 3D localization/quantification	Antibody penetration; signal-to-noise impacts sensitivity; protocol-dependent	Clearing protocol; imaging settings; segmentation method; validation modality	[14]
scRNA-seq/scATAC-seq	Unbiased state discovery; regulatory inference	Dropout/dissociation bias affects sensitivity; snapshot→trajectory inference	Cell numbers; QC metrics; batch correction; signature definitions	[37,92,124]
Spatial transcriptomics	Niche localization of putative transitions	Resolution limits; transcript capture affects sensitivity; snapshot inference	Platform/resolution; tissue handling; normalization; validation markers	[91,93]

Note: Sensitivity/specificity refer to the confidence of EndMT calls; the symbol “→” denotes a directional transition of inference; transient/partial states are often rare and time-dependent; reproducibility requires standardized reporting of protocols, QC metrics, and EndMT signature/threshold definitions.

**Table 3 ijms-26-11724-t003:** EndMT-targeted interventions.

Strategy/Mechanism	Representative Agents	Model & Readouts	Effect	References
Pathway inhibitors:TβR1/canonical SMAD	SB-431542, A-83-01; Galunisertib	ECs; MI/IPF/PAH models; marker panel	EndMT markers↓, migration↓; fibrosis burden↓	[8,28,32,145,146,147,148,160]
Non-canonical:Rho/ROCK, FAK, PI3K/AKT, MAPK	ROCK/FAK inhibitors	Atheroprone-flow ECs; lung vascular models	Stress fiber↓, mesenchymal program↓	[54,55,56]
Epigenetic:HDAC/DNMT/EZH2/JMJD2B	HDAC inhibitors; 5-aza-2′-deoxycytidine	In vitro ECs; fibrosis models	Chromatin re-opening→EndMT↓	[22,23,24,49,53,55,56]
miRNA therapeutics:restore anti-EndMT/block pro-miRs	miR-200b mimic; anti-miR-21 (LNP/exosome)	ECs, iPSC-ECs; injury models	ZEB1/2↓; EndMT↓; maintain EC identity	[47,58,63,84,157,158,159]
Biologics:anti-TGF-β/endoglinaxis	mAbs/ligand traps (context-dependent)	Fibrosis/PAH models	SMAD-driven EndMT↓	[8,17,18,160,162]
Cell/Gene:iPSC-ECs; CRISPR edits	SNAI/ZEB KO; miR-cassettes KI	Graft stability assays; in vivo repair	EndMT resistance↑, preserve function	[72,128,129,130,131,132,133,138,139,170,171,172,173]
Natural products/small molecules:multi-pathway	Resveratrol; Curcumin	EC EndMT assays; fibrosis models	ROS/NF-κB↓;EndMT↓	[168,169]

Note: Arrows indicate direction of change relative to baseline ECs (↑ up-regulated/increased; ↓ down-regulated/decreased).

## Data Availability

No new data were created or analyzed in this study. Data sharing is not applicable to this article.

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
