# Peer review of "Endothelial-to-Mesenchymal Transition in Health and Disease: Molecular Insights and Therapeutic Implications"

_ijms, 2025, doi:10.3390/ijms262311724_

Round 1

Reviewer 1 Report

Comments and Suggestions for Authors

In the present review, Ran Kim and Woochul Chang discussed about Endothelial-to-Mesenchymal Transition in Health and Disease: Molecular Insights and Therapeutic Implications. The authors described the End-MT process with the different pathways implicated. They have also referring to the pathological contexts in which this process is present/exacerbated and described the methods for End-Mt assessment (in vitro and in vivo) as well as therapeutic strategies for its inhibition. The review is very interesting and well structured. However, it should be improved making some minor changes.

  1. In the section “Molecular mechanisms and pathophysiological roles of EndMT” the authors described the different pathways involved in End-Mt process. Wnt/ b catenin is a key regulator of End-MT (PMID: 36001597; 29761409; PMID: 40180907). The authors cited this pathway only in the introduction and in the Table 1. Given the importance of this pathway in the End-MT, it would be better to insert a paragraph (as already done for Transforming growth factor-β) also for Wnt/ b catenin in the section "Molecular mechanisms and pathophysiological roles of EndMT". Moreover, it has been reported a cross-talk between TGFb signaling and for Wnt/ b catenin pathway in End- MT (see PMID: 32373613), so the authors should underline it in the manuscript.
  2. In the section “Molecular mechanisms and pathophysiological roles of EndMT”, paragraph 2.3.3. MicroRNA, the authors indicated miR-200b as inhibitor of End-MT. In the family of miR-200, miR-200c-3p, unlike miR-200b, induces End-Mt, as reported (PMID: 33125708; PMID: 34525946). Moreover, miR-632 has been also reported as activator of End-Mt and EMT and linked to TGFb signalling. The authors could add those two miRNAs as drivers of End-MT in the paragraph 2.3.3 microRNA.

3. Table 1 line 1: TGFb is reported as activator of End-Mt in different disease context (Cardiac, renal, and pulmonary fibrosis). TGFb is reported as an inductor of End-Mt also in cardiovascular disease (Cardiac fibrosis, aortic aneurysms, etc..), so the authors should add these disease context in the TGFb line.

Author Response

1. In the section “Molecular mechanisms and pathophysiological roles of EndMT” the authors described the different pathways involved in EndMt process. Wnt/β-catenin is a key regulator of EndMT (PMID: 36001597; 29761409; PMID: 40180907). The authors cited this pathway only in the introduction and in the Table 1. Given the importance of this pathway in the EndMT, it would be better to insert a paragraph (as already done for Transforming growth factor-β) also for Wnt/β-catenin in the section "Molecular mechanisms and pathophysiological roles of EndMT". Moreover, it has been reported a cross-talk between TGF-β signaling and for Wnt/β-catenin pathway in EndMT (see PMID: 32373613), so the authors should underline it in the manuscript.

A: In response to the reviewer’s recommendation, we added a new subsection summarizing Wnt/β-catenin signaling as a key regulatory axis in EndMT and its functional interplay with TGF-β signaling. The revised manuscript includes this content in Section 2.2. Wnt/β-catenin signaling in EndMT, along with corresponding references. The revised text is highlighted in red in the manuscript (Page 3, Lines 103-115).

2. In the section “Molecular mechanisms and pathophysiological roles of EndMT”, paragraph 2.3.3. MicroRNA, the authors indicated miR-200b as inhibitor of EndMT. In the family of miR-200, miR-200c-3p, unlike miR-200b, induces EndMT, as reported (PMID: 33125708; PMID: 34525946). Moreover, miR-632 has been also reported as activator of EndMT and EMT and linked to TGF-β signaling. The authors could add those two miRNAs as drivers of EndMT in the paragraph 2.3.3 microRNA.

A: In response to the reviewer’s recommendation, we expanded Section 2.4.3 MicroRNA to more explicitly describe the context-dependent roles of miRs in EndMT, including emerging evidence that certain miRs can promote EndMT. The revised manuscript includes these additions with the corresponding references. The revised text is highlighted in red in the manuscript (Page 4, Lines 166-168, 173-175).

3. Table 1 line 1: TGF-β is reported as activator of End-Mt in different disease context (Cardiac, renal, and pulmonary fibrosis). TGF-β is reported as an inductor of EndMT also in cardiovascular disease (Cardiac fibrosis, aortic aneurysms, etc..), so the authors should add these disease context in the TGF-β line.

A: In response to the reviewer’s suggestion, we revised Table 1 to expand the disease contexts associated with TGF-β-induced EndMT. Specifically, we added cardiovascular disease contexts, including cardiac fibrosis/valvular disease and aortic aneurysms, in addition to renal and pulmonary fibrosis. The revised text is highlighted in red in the manuscript (Table 1, TGF-β row).

Reviewer 2 Report

Comments and Suggestions for Authors

In this review, the authors focus on endothelial-to-mesenchymal transition (EndMT), a cellular program implicated in fibrosis, vascular remodeling, and the tumor microenvironment across multiple organs. The mechanistic pathways involved are synthesized, encompassing pathologies affecting various organ systems. The review also includes different approaches for EndMT detection, some of which enable delineation of transitional states, support inferences regarding process reversibility, and localize EndMT within specific tissue niches. Finally, the authors summarize therapeutic agents and propose a translational roadmap that integrates patient-derived platforms, such as iPSC/organoid and organ-on-a-chip systems, to stratify EndMT states and prioritize therapeutic targets.

Comments:

Compared with previously published material, this review:

-Provides an updated synthesis of canonical and non-canonical signaling pathways.

-Highlights cutting-edge technologies, including spatial transcriptomics and 3D tissue clearing, which are not yet consistently covered in the literature.

-Bridges basic mechanisms with translational considerations, proposing a roadmap for target prioritization—an element that expands the utility of the review for both researchers and clinicians.

These aspects collectively enhance the field’s conceptual clarity and may guide future experimental design.

Methodological improvements:

-Clarify inclusion/exclusion criteria for the literature incorporated, to help readers assess scope and potential biases.

-Differentiate more explicitly between in vitro, in vivo, and ex vivo evidence, especially when discussing reversibility of EndMT and transitional cellular states.

-Provide a critical appraisal of emerging technologies: their sensitivity, specificity, limitations in detecting transient phenotypes, and reproducibility across laboratories.

-Consider adding a table summarizing the strengths and limitations of key methodological platforms.

The conclusions are broadly consistent with the mechanistic and methodological evidence presented. However, they could be strengthened by:

-More explicitly summarizing the major unresolved questions in EndMT biology.

-Highlighting challenges in translating in vitro or organoid findings to human physiology.

-Discussing caveats related to therapeutic targeting, including pathway redundancy, context dependency, and potential off-target effects.

Author Response

- The conclusions could further emphasize the remaining challenges and future perspectives in the field. Additionally, the authors might consider addressing some of the limitations of the emerging techniques and therapeutic targets.

A: In response to the reviewer’s suggestion, we revised the Conclusions section throughout to more clearly highlight the remaining challenges and future perspectives in the field, including key limitations of emerging techniques and therapeutic targets. These changes are reflected in the revised manuscript (Page 15).

-Clarify inclusion/exclusion criteria for the literature incorporated, to help readers assess scope and potential biases.

A: In response to the reviewer’s suggestion, we added a new subsection, Section 1.1. Scope and literature selection, to clarify the scope of the review and our literature inclusion/exclusion criteria. The revised text is highlighted in blue in the manuscript (Page 2, Lines 52-70).

-Differentiate more explicitly between in vitro, in vivo, and ex vivo evidence, especially when discussing reversibility of EndMT and transitional cellular states.

A: In response to the reviewer’s suggestion, we revised the manuscript throughout to more explicitly differentiate in vitro, in vivo, and ex vivo/human evidence, particularly in sections discussing EndMT reversibility and transitional cellular states. The revised text is highlighted in blue in the manuscript.

-Provide a critical appraisal of emerging technologies: their sensitivity, specificity, limitations in detecting transient phenotypes, and reproducibility across laboratories.

A: In response to the reviewer’s suggestion, we added brief critical appraisals of emerging technologies, addressing sensitivity, specificity, detection of transient transient phenotypes, and inter-laboratory reproducibility, at the end of the relevant existing paragraphs in Section 4.1.3, 4.3.1, 4.3.2, and the Conclusion. The revised text is highlighted in blue in the manuscript (Page 9, Lines 350-354; Page 10, Lines 391-398; Page 10-11, Lines 412-419)

-Consider adding a table summarizing the strengths and limitations of key methodological platforms.

A: In response to the reviewer’s suggestion, we added a new table (Table 2) summarizing the strengths and limitations of key methodological platforms used to study EndMT, including major considerations for interpretation and reproducibility.

-More explicitly summarizing the major unresolved questions in EndMT biology.

A: In response to the reviewer’s suggestion, we revised the Conclusion section to more explicitly summarize the major unresolved questions in EndMT biology, and incorporated these additions at the appropriate place in the Conclusion. The revised text is highlighted in blue in the manuscript (Page 15, Lines 566-570).

-Highlighting challenges in translating in vitro or organoid findings to human physiology.

A: In response to the reviewer’s suggestion, we added a concise statement in the Conclusion to underscore the key challenges in translating findings from in vitro and organoid/on-chip models to human physiology brief critical appraisal statement within each subsection. The revised text is highlighted in blue in the manuscript (Page 15, Lines 549-553).

-Discussing caveats related to therapeutic targeting, including pathway redundancy, context dependency, and potential off-target effects.

A: In response to the reviewer’s suggestion, we expanded the relevant passage in the Conclusions to address key caveats of therapeutic targeting in EndMT, including pathway redundancy, context dependency, and potential off-target effects. The revised text is highlighted in blue in the manuscript (Page 15, 574-579).

Reviewer 3 Report

Comments and Suggestions for Authors

Dear authors,

Thank you for providing this manuscript, which presents an engaging analysis of over 150 scientific studies on the topic of endothelial-to-mesenchymal transition (EndMT). The paper is written concisely and clearly, making it easy to understand.

Nevertheless, in my opinion, to enhance the comprehension and value of the article, it is worth considering the possibility of conceptually combining the chapters. In particular, it would be useful to further explain the characteristics of organ pathologies associated with EndMT in the context of the involvement of various molecular mechanisms.

I also request that you clarify which disease models and assessment methods are most suitable or popular for studying specific organ pathologies associated with EndMT. This addition would significantly enrich the material and make it more informative for readers.

Author Response

Nevertheless, in my opinion, to enhance the comprehension and value of the article, it is worth considering the possibility of conceptually combining the chapters. In particular, it would be useful to further explain the characteristics of organ pathologies associated with EndMT in the context of the involvement of various molecular mechanisms.

A: We thank the reviewer for this suggestion. Accordingly, we added a short bridge paragraph at the end of Section 2 to conceptually connect the mechanistic framework to organ pathologies, updated each organ subsection in Section 3 with brief mechanism-summary sentences linking EndMT-related molecular pathway to organ-specific disease features. The revised text is highlighted in green in the manuscript (Page 5, Lines 193-201; Section 3).

I also request that you clarify which disease models and assessment methods are most suitable or popular for studying specific organ pathologies associated with EndMT. This addition would significantly enrich the material and make it more informative for readers.

A: We thank the reviewer for this helpful suggestion. Accordingly, we added a new subsection (Section 4.4) that summarizes organ-specific EndMT-associated pathologies and the corresponding assessment/readout methods. The revised text is highlighted in green in the manuscript (Page 11, Lines 420-434).
